# The Biological Role of Platelet Derivatives in Regenerative Aesthetics

**DOI:** 10.3390/ijms25115604

**Published:** 2024-05-21

**Authors:** Lorena Cristina Santos, Giselle Lobo Lana, Gabriel Silva Santos, Silvia Beatriz Coutinho Visoni, Rayssa Junqueira Brigagão, Napoliane Santos, Rafaela Sobreiro, Andreza da Cruz Silva Reis, Bruno Lima Rodrigues, Sabrina Ferrari, Claudia Herrera Tambeli, José Fábio Lana

**Affiliations:** 1Biomedical Science, Hospital das Clínicas de Goiás, Goiânia 74605-020, Brazil; lorenacsantos@yahoo.com.br; 2Orthopedics, Brazilian Institute of Regenerative Medicine (BIRM), Indaiatuba 13334-170, Brazil; giselleclobo@gmail.com (G.L.L.); visonisilvia@gmail.com (S.B.C.V.); rayssajbrigagao@yahoo.com.br (R.J.B.); dranapolianesantos@gmail.com (N.S.); ipc.sobreiro@gmail.com (R.S.); andrezacruzenf@gmail.com (A.d.C.S.R.); brunolr.ioc@gmail.com (B.L.R.); sabrinaferrariart@gmail.com (S.F.); josefabiolana@gmail.com (J.F.L.); 3Institute of Biology, State University of Campinas (UNICAMP), Campinas 13083-862, Brazil; 4Medical School, Max Planck University Center (UniMAX), Indaiatuba 13343-060, Brazil; 5Regenerative Medicine, Orthoregen International Course, Indaiatuba 13334-170, Brazil; 6Clinical Research, Anna Vitória Lana Institute (IAVL), Indaiatuba 13334-170, Brazil

**Keywords:** platelet-rich plasma, platelet-rich fibrin, growth factors, tissue regeneration, aesthetic medicine

## Abstract

Bioproducts derived from platelets have been extensively used across various medical fields, with a recent notable surge in their application in dermatology and aesthetic procedures. These products, such as platelet-rich plasma (PRP) and platelet-rich fibrin (PRF), play crucial roles in inducing blood vessel proliferation through growth factors derived from peripheral blood. PRP and PRF, in particular, facilitate fibrin polymerization, creating a robust structure that serves as a reservoir for numerous growth factors. These factors contribute to tissue regeneration by promoting cell proliferation, differentiation, and migration and collagen/elastin production. Aesthetic medicine harnesses these effects for diverse purposes, including hair restoration, scar treatment, striae management, and wound healing. Furthermore, these biological products can act as adjuvants with other treatment modalities, such as laser therapy, radiofrequency, and microneedling. This review synthesizes the existing evidence, offering insights into the applications and benefits of biological products in aesthetic medicine.

## 1. Introduction

Biological products are diverse substances, including vaccines, growth factors, immunomodulators, monoclonal antibodies, and hematological components. Various studies have demonstrated the use of numerous biologics in almost every field of medicine. The use of autologous hematological components, especially platelet-rich plasma (PRP), has become a highly attractive therapeutic tool for various applications since the biological functions of these products go beyond hemostasis [1].

According to the International Olympic Committee, PRP is an autologous preparation derived from whole blood in which platelets are concentrated in a small fraction of the plasma [2] (Figure 1).

Platelet concentrates carry various growth factors contained in alpha and dense granules. Alpha granules possess seven important growth factors: platelet-derived growth factors (PDGFaa, PDGFbb, and PDGFab), transforming growth factor beta (TGFβ1 and 2), epithelial growth factor (EGF), and vascular endothelial growth factor (VEGF) [1]. These growth factors regulate processes such as cell proliferation, matrix remodeling, differentiation, angiogenesis, and chemotaxis. Dense granules contain bioactive substances such as ADP, ATP, serotonin, and calcium, which, after activation and subsequent platelet degranulation, increase membrane permeability and stimulate regenerative processes, promoting vascular remodeling and immunomodulation through the release of signaling molecules [1,2,3,4]. In addition to these advantages, since platelet-derived bioproducts are autologous products, they do not present the risk of allergic induction and have been extensively used in various medical specialties, including cardiac surgery, oral surgery, orthopedics, sports medicine, and facial plastic surgery. Furthermore, these bioproducts have undergone a significant upsurge in dermatology and aesthetic procedures in recent years [5,6,7]. These biological products have served as regenerative agents for several years, leveraging growth factors derived from peripheral blood to induce vascularization in different tissues. Upon activation, these cells form a fibrin network in a liquid state, releasing a plethora of growth factors that foster tissue regeneration by stimulating cell proliferation, differentiation, and migration and the production of collagen and elastin [8]. These effects have been explored in aesthetic medicine for different purposes, such as alopecia treatment, scar treatment, surgery, and wound healing (Figure 2). These products may also be used as adjuvants in other treatment modalities, including ablative and nonablative laser therapy, bipolar radiofrequency, and microneedling [9,10,11,12,13,14].

Currently, there are different types of PRP preparation methods for which parameters such as the number of centrifugation rounds, relative centrifugal force (RCF), and time can vary. This influences the platelet integrity, composition, and effectiveness of the product. Additionally, the variance among the methods and contents between PRP products results in different terminologies and results, making standardization and reproducibility challenging [3]. For instance, studies have reported that the anticoagulants used in PRP preparations may also interfere with several processes, such as wound healing [8]. For these reasons, platelet-rich fibrin (PRF), an alternative platelet concentrate, was developed with no additives (anticoagulants) and is produced under lower centrifugal forces (Figure 3). This protocol enhances the content and distribution of the cells and growth factors within PRF matrices. Furthermore, it enables injection in a manner similar to PRP, offering the added benefits of fibrin clot formation shortly after injection into the target tissue. Its feasibility is attributed to its easy handling and a growth factor content relatively comparable to those of certain PRP products [15].

Considering these findings, when new techniques are used, PRP and its derivatives can deliver results that meet patient and physician expectations.

Therefore, the objective of this review was to highlight the roles of PRP and its derivatives, including PRF, in aesthetics. This manuscript will also cover their associated benefits when compared to or associated with conventional treatments.

## 2. Methods

A thorough literature review was conducted to offer a comprehensive understanding of the emerging applications of PRP and its derivatives in aesthetic medicine and dermatology. The PUBMED database was employed from October to December 2023 to identify relevant reports. The search strategy involved combining the terms “aesthetic use” with variations of platelet-rich plasma (PRP), including platelet-rich fibrin (PRF), injectable platelet-rich fibrin (i-PRF), and PRP itself. This search yielded approximately 700 results (476 for PRP and 210 for PRF). Titles, abstracts, and full texts were meticulously screened and selected by the authors. A total of 69 articles were included in this study based on their relevance and publication years, of which 43 were randomized clinical trials, as listed in Table 1.

## 3. Aesthetic Conditions

### 3.1. Alopecia

Currently, alternatives for treating hair loss focus on promoting cell proliferation and differentiation during the growth cycle. The effective pharmacological treatments are 2–5% topical minoxidil, which works through several mechanisms, including arteriolar vasodilation, anti-inflammatory effects, Wingless and Int-1 gene (Wnt)/β-catenin signaling, and oral finasteride, which is a selective 5α-reductase type II inhibitor. Although these drugs improve patients with androgenetic alopecia (AGA), researchers continue to search for more effective alternatives and limited side effects. For these reasons, due to the significant ability of PRP to promote tissue regeneration, new treatment protocols have been developed using this bioproduct, demonstrating efficacy in the treatment of alopecia, especially in AGA [57].

Although PRP is widely and effectively used to treat hair loss, the exact mechanism of action of PRP for this purpose has yet to be fully elucidated [58]. It is known that the hair follicle is a self-renewing mini-organ that undergoes metabolic and morphological changes during its cycle. It involves the anagen hair growth phase, the catagen hair growth phase, the return phase of the cycle, the telogen hair growth phase, and the inactive phase. PRP can have various effects on hair, and one of the key factors is its antiapoptotic effect, which is activated by the Bcl-2 protein and Akt signaling. This contributes to hair growth by increasing the longevity of dermal papilla cells throughout the hair cycle. Additionally, research has suggested that PRP treatment upregulates the FGF-7/β-catenin signaling pathway, leading to follicular stem cell differentiation, stimulating hair growth, and prolonging the anagen phase of the growth cycle. PRP also appears to enhance the perifollicular vascular plexus, elevating angiogenic factors such as VEGF and PDGF [18].

A study on AGA treatment with PRP in rats showed that this biologic product accelerated the hair cycle, suggesting that PRP injections could promote hair growth. To verify the therapeutic efficacy of PRP in humans, researchers conducted a randomized, placebo-controlled, double-blinded study with 52 patients with AGA. The study results demonstrated a substantial increase in the hair density after PRP injections within a short timeframe. At the 6-month mark, the PRP group exhibited significant improvements in their hair counts, hair diameters, and proportion of anagen hairs compared to the control group and baseline measurements [13].

Multiple clinical studies provide evidence supporting the efficacy and safety of using PRP for treating AGA in men. In a randomized, placebo-controlled study involving 23 male patients, hair growth parameters were assessed three months after the initial PRP session and compared with the baseline values in the treated and control areas. The study revealed significant increases in the average hair count and total hair density in the PRP-treated area. Microscopic evaluation two weeks after the last PRP treatment indicated increases in the epidermal thickness in the capillary skin compared to the baseline, accompanied by a rise in the number of follicles. Additionally, researchers observed an increase in basal epidermal keratinocytes and follicular bulb cells [18]. These data showed that PRP can be an effective treatment option for hair loss.

Butt and colleagues [21] evaluated the effectiveness of PRP for hair restoration in 30 patients, including men and women, using various parameters, such as the terminal-to-vellus hair ratio, the hair density, the hair-pulling test, photographs, the physician global assessment score, and the patient global assessment score. The authors showed that PRP is an effective treatment option for androgenetic alopecia, resulting in greater hair density and physician and patient global assessment scores and an increase in the terminal-to-vellus hair ratio.

The type of study design directly influences the results. Several split-scalp studies evaluating intradermal PRP versus saline have shown that PRP improves androgenetic alopecia, but other studies have failed to prove this. Gentile and collaborators [20] performed a randomized, placebo-controlled trial comparing the hair regrowth of patients receiving PRP injections via controlled, programmed mechanical injections at a depth of 5 mm using a medical injection gun versus placebo injections. Patients underwent three sessions, resulting in an increase of 33.6 hairs in the target area and an average increase in density of 45.9 hairs per cm^3^ compared to those in the placebo group. Similarly, Alves and Grimalt [19] conducted a study on twenty-five patients with AGA. They revealed that PRP treatment has a positive effect on AGA and could even be regarded as a promising therapy for this disorder.

Conversely, Shapiro et al. [23] analyzed the effect of the intradermal injection of PRP on hair regrowth and thickness compared with saline. While the hair density significantly increased in the PRP-treated area compared to the baseline, there was also a modest increase in the placebo-treated areas. These findings suggest no significant difference in the hair density changes between the two groups. The variations in the study outcomes may be attributed to differences in the techniques and types of injections used, influencing the diffusion and action of PRP. This “design effect” could explain the inconsistent findings in PRP studies, highlighting the need for additional research to identify factors affecting treatment outcomes.

Most AGA studies are carried out on men or on groups containing men and women; however, the effects in these two groups have been yet to be evaluated separately. According to a meta-analysis assessing the efficacy of PRP separately in men and women, PRP significantly increased the hair diameter in both sexes, but the hair density was significantly increased only in men [59]. Although the efficacy of PRP in men and women differed between the two groups, the administration of PRP was promising in both.

Numerous techniques, including PRP therapy and microneedling, have been employed individually or in combination to promote hair growth and mitigate hair loss. Microneedling is based on the premise that limited and controlled injury to the skin triggers a natural healing cascade, provoking the release of growth factors and, therefore, stimulating the vital dermal structures necessary for hair growth and strengthening. With this in mind, Muhammad and colleagues compared the efficacies of the combined or isolated application of PRP and microneedling in treating AGA. The group that underwent PRP application through microneedling experienced a significantly greater average hair count than the group that received PRP alone. Compared with PRP alone, microneedling + topical PRP resulted in a greater increase in the hair count in individuals experiencing hair loss. Combining these two techniques demonstrated synergistic effects and was proven to be more effective than either method alone, especially in patients with AGA [25].

A study published in 2021 demonstrated the effectiveness of PRP treatment compared to intradermal application in AGA patients treated with microneedling. The hair count, hair density, terminal hair count, and terminal hair density significantly differed between the two groups and before treatment. A statistically significant difference was found between the averages of anagen and telogen hair in addition to the hair length in the groups treated with microneedling [26].

Several authors have combined PRP with other substances to improve the treatment efficacy of AGA. In a recent study, Wu and collaborators [27] used PRP plus fibroblast growth factor (PRPF) and minoxidil. In this prospective, randomized, controlled study, 75 patients with AGA were divided into three groups according to the type of treatment used. In group 1, PRPF was used as a monotherapy through intradermal injections; in group 2, patients received only 5% topical minoxidil twice a day; in group 3, PRPF injections were performed in combination with topical applications of minoxidil. All patients treated with PRPF showed improvements in their hair counts, terminal hair counts, and growth rates compared to those treated with minoxidil monotherapy.

Wei et al. [28] also compared the efficacy and safety of PRP combined with topical minoxidil for treating AGA. Thirty male patients were divided into two treatment groups: the first group received PRP with 5% topical minoxidil, and the second group received PRP with a topical placebo. The hair density/quantity, clinical efficacy, and patient satisfaction were more pronounced in the first group. This study showed that the effects of PRP and minoxidil treatment exceeded those of PRP alone, demonstrating the former as a potentially beneficial treatment strategy for AGA.

Although the application of PRP in hair loss conditions is mainly based on AGA, some studies have reported its benefits in other types of alopecia, such as telogen effluvium. Chronic telogen effluvium is characterized by diffuse scalp hair loss. El-Dawla et al. evaluated the safety and efficacy of different preparation methods of PRP versus a placebo in thirty women with chronic telogen effluvium. They concluded that PRP can be acknowledged as an effective therapeutic alternative for patients with chronic telogen effluvium [29]. Moftah and colleagues [24] studied different PRP preparation protocols regarding the number of spins, centrifugal force, type of centrifuge, and tube size to evaluate which protocol has better clinical efficacy in treating female pattern hair loss (FPHL). Patients were divided into four groups as follows: one spin using a centrifugation force of 1000× *g* for 10 min; two spins using a centrifugation force of 250× *g* for the first and 450× *g* for the second, both for 10 min; two spins with a centrifugation force of 250× *g* for the first and 1000× *g* for the second, both for 10 min. All of these groups used small sodium citrate tubes (2 mL VACUTEST tubes, Buffered Citrate 9NC 3.2%, made in Italy) and a non-digital centrifuge (Electronic Centrifuge 80-1; maximum speed: 4000 rpm; timer range: 0–60 min; capacity: 20 mL × 6; made in China). In the last group, PRP was prepared with one spin, using a centrifugation force of 220× *g* for 10 min, with a large sodium citrate tube (9 mL VACUETTE tube, 9NC coagulation sodium citrate 3.2%, made in Austria) and a digital centrifuge (HERMLE z 326 k centrifuge; maximum speed: 18,000 rpm; maximum capacity: 4 × 100 mL; timer: from 10 s to 99 min; temperature range: from −20 to 40 °C; made in Germany).

This study revealed that the PLT was lower in groups I, II, and III, in which a non-digital centrifuge was used at a high rotation speed with small sodium citrate tubes. Conversely, protocols using a digital centrifuge, large sodium citrate tubes, and a low-speed spin (220× *g*) were more effective at preparing PRP. These findings support existing research suggesting an inverse relationship between the platelet count and centrifugation speed. In terms of efficacy, a study revealed that PRP is effective and safe for treating female pattern hair loss (FPHL), as all patients exhibited statistically significant increases in their percentages of terminal hairs and average hair widths after treatment [24].

To investigate whether the discrepancies in the results reported in the literature regarding PRP in patients with androgenetic alopecia could be linked to the protocol used and correlated with the platelet and growth factor levels, Rodrigues et al. conducted an analysis of the platelet count and growth factor levels in PRP and their correlation with hair growth. In this study, 26 patients were randomly assigned to receive four subcutaneous injections of either PRP or saline. The authors observed the favorable use of PRP as a therapeutic alternative for treating androgenetic alopecia, but no association was found between the platelet count and the evaluated growth factors. The conclusion was that clinical improvement might be associated with other mechanisms [22].

Rinaldi and collaborators compared the efficacy and safety of PRP with those of triamcinolone acetonide (TrA) or a placebo in 45 patients with alopecia areata (A.A.). Patients who underwent PRP treatment exhibited significantly greater hair regrowth than those treated with the TrA or placebo. The study revealed that 38% and 71% of patients in the TrA group experienced regression at the 6th and 12th months, respectively. In contrast, no patients in the PRP group experienced recurrence at 6 months, and only 31% did so at 12 months. Overall, 96% of patients in the PRP group achieved the regrowth of fully pigmented hair from the initiation of hair growth. In contrast, only 25% of patients in the TrA group had pigmented hair at the beginning of hair growth [16].

Furthermore, Vazques et al. [17] evaluated PRF for the treatment of A.A. in a 28-year-old patient who developed this condition after a symptomatic COVID-19 infection. He underwent two PRF intradermal sessions, the first in March and the second in May, at which point his A.A. resolved at the 6-month follow-up. The authors concluded that PRF for A.A. is a promising treatment for patients with this autoimmune disease.

PRP and PRF, used alone or in combination with various protocols, techniques, or other agents, such as 5% minoxidil, appear promising for treating different types of alopecia or hair loss in both men and women. However, factors such as the study design, PRP preparation protocols, nature and severity of the disease, PRP diffusivity, and potential associations with other substances can impact the quality of the results. Consequently, further research is needed to establish a standard protocol that optimizes treatment outcomes.

### 3.2. Skin Rejuvenation

The demand for aesthetic procedures, particularly rejuvenation, has steadily increased in recent years due to the aging global population. Skin aging is influenced by intrinsic factors, such as genetic background and chronological age, as well as extrinsic factors, such as exposure to ultraviolet radiation, trauma, air pollution, alcohol consumption, nutritional issues, smoking, and reactive oxygen species causing DNA damage. These factors contribute to decreased or hyperexpression in melanocytes, reduced fibroblast activity, and diminished collagen and elastin synthesis, leading to dermal disruption and the dysregulation of the stem cell population responsible for tissue repair [60,61]. Subsequent skin changes involve hyper- or hypopigmentation, skin laxity, and the development of both superficial and deep wrinkles, significantly impacting self-image and social acceptance [62]. Regarding rejuvenation, the primary goal is to counteract or minimize the aging process using either surgical or noninvasive methods. As the demand for aesthetic procedures to prevent related changes and enhance skin quality has risen, there has been a notable decline in the number of patients opting for surgical interventions [61]. Among the nonsurgical aesthetic procedures, PRP has been extensively researched and has yielded favorable results with high patient satisfaction and no significant adverse effects [62]. PRP offers numerous benefits in facial rejuvenation, addressing concerns such as atrophic acne scars, pigmentation disorders, wrinkles, folds, loss of elasticity, and tissue volume loss. The exact mechanism of action of PRP and its derivatives in facial rejuvenation has yet to be fully understood. It is hypothesized that the growth factors in PRP may facilitate tissue repair, increase cell proliferation, and influence the expression of differentiation genes. This, in turn, supports angiogenesis and cellular rejuvenation processes, potentially leading to more enduring effects than other procedures [63]. In vitro studies have indicated that PRP may enhance collagen expression, remodel the extracellular matrix, promote fibroblast proliferation, and facilitate fibroblast differentiation into myofibroblasts [64].

The application of PRP and its derivatives is intended to improve the skin quality, texture, and tone via injection or topical application combined with microneedling either individually or in combination with other aesthetic procedures. Banihashemi and colleagues [38] tested two sessions of pure PRP with a 3-month interval for facial rejuvenation in 30 female participants. According to patient reports, they observed significant improvements in periorbital dark circles and nasolabial folds. They concluded that facial rejuvenation with PRP is an effective and noninvasive technique.

Among the different types of wrinkles, periorbital wrinkles are the most common and challenging to rejuvenate, as the periorbital skin is thinner. Considering that PRP and plasma gel have been used for skin rejuvenation, Diab and colleagues compared the use of both products for periorbital wrinkles in 40 female patients. The PRP and plasma gel were injected into each side of the face during two treatment sessions four weeks apart. Patients were followed up 2 weeks after each treatment session and 12 weeks after the last session. The authors showed that after the second session, both modalities had significantly improved periorbital wrinkles, achieving superior results with the plasma gel. However, they were not able to ameliorate periorbital hyperpigmentation [41]. However, Nilforoushzadeh et al. showed that the combination of an erbium-doped yttrium aluminum garnet (Er: YAG) laser and PRP is significantly more effective for periorbital hyperpigmentation and wrinkles than Er: YAG laser monotherapy [39].

Similarly, in a prospective clinical trial, Mahmoodabadi and colleagues verified the effectiveness of PRP and its derivatives in treating periorbital wrinkles [45]. PRF was injected into the periorbital regions of 15 volunteers. The results demonstrated improvements in deep, delicate, and small wrinkles, periorbital hyperpigmentation, and the overall freshness of the skin at the injection site. Participants reported swelling at the injection site lasting up to one day after the procedure, which resolved on its own without any complications.

Based on the evidence that PRP conveys favorable results for hyperpigmentation treatment, Gonzalez-Ojeda and colleagues [43] studied the role of PRP in treating melasma. Twenty female patients with melasma underwent three intradermal PRP sessions at an interval of 15 days, which were evaluated before and after treatment. Comparisons were made regarding the concentration of melanin in the treated area, the severity index score, the degree of patient satisfaction, and the degree of histological changes. Through dermatoscopy examinations, decreases in pigmentation after treatment with PRP and histopathological improvements were reported. Reductions in skin atrophy, solar elastosis, and inflammatory infiltration were also observed.

Several cosmetics and chemical peels, including tranexamic acid, constitute the therapeutic armamentarium for treating melasma with some success. Therefore, Patil and Bubna [42] compared the benefits and safety of tranexamic acid (4 mg/mL) versus PRP for melasma treatment. In total, 40 patients with melasma were randomly distributed into two groups and received either tranexamic acid or PRP. Patients received weekly intradermal injections for one month, totaling five injections. No significant adverse effects were found in either group, and both had rapid or substantial improvements in melasma, demonstrating that these agents are effective and safe therapeutic options for this disease.

Mumtaz et al. compared the effectiveness of PRP to that of tranexamic acid (4 mg/mL) in the treatment of melasma, obtaining superior results with PRP. The authors divided 64 patients with melasma into two groups: one with 1 mL of PRP and one with intradermal tranexamic acid. Treatment was offered every four weeks for twelve weeks, and the final result was observed in the 24th week [40].

Since PRP has potential benefits for facial skin rejuvenation, it has been used in several investigations. A study examined the effectiveness of PRP intradermal injections for facial rejuvenation through biometric parameters and patient satisfaction. Significant reductions in the number and area of blemishes, decreases in the count and depth of wrinkles, and improvements in the redness and firmness of the skin were observed. Regarding the patient satisfaction index, after six months, an average score greater than 90% was achieved [33]. A similar study evaluated the efficacy and safety of intradermal PRP injections for rejuvenation and wrinkle treatment in 20 patients with different types of wrinkles. In this investigation, the intradermal injection of PRP was well tolerated and capable of rejuvenating the face, promoting the significant correction of wrinkles, especially in the nasolabial folds [30].

When combined with other skin-boosting biomolecules, such as hyaluronic acid, PRP has emerged as an effective treatment for enhancing the skin quality, addressing bleeding, and combating signs of aging. An extensive study involving 80 patients with facial aging examined the clinical efficacy, adverse reactions, and durability of PRP + skin booster effects. Over a year of observation, patients reported positive changes in their skin condition, enhanced quality of life, and increased satisfaction with their appearance following treatment. The study concluded that PRP + skin booster was effective and safe at alleviating issues such as coarse pores and wrinkles, contributing to overall facial rejuvenation. These findings offer compelling evidence supporting the clinical application of PRP in skin booster treatments [44].

Hassan et al. focused on testing injectable platelet-rich fibrin (i-PRF) for facial skin rejuvenation. The study reported significant improvements in various skin parameters, including surface spots, pores, wrinkles, ultraviolet spots, skin texture, and porphyrins, at the 3-month follow-up. Patient satisfaction with appearance also showed substantial enhancements, particularly regarding skin quality and overall facial appearance, covering areas such as the cheeks, lower face, jawline, and lips. Notably, the study highlighted the safety and effectiveness of intradermal i-PRF, emphasizing increased patient satisfaction without significant adverse effects [34]. In contrast, Silva and collaborators used lyophilized PRP because the facility needed to obtain numerous samples via single venipuncture, which is helpful for multiple injections. The authors evaluated the effect of this preparation by monthly intradermal injections compared with saline in treating skin aging through a phase II pilot study. They found that applying freeze-dried PRP via mesotherapy did not improve skin aging [37]. A split-face study comparing the benefits of PRP on photoaged skin with those of saline control skin in male and female participants with bilateral cheek rhytids was performed by Alam and colleagues. The average photoaging scores related to fine lines, mottled pigmentation, roughness, and pallor, reported by two dermatologists, showed no significant differences between the PRP and control groups. However, patient satisfaction regarding texture, wrinkles, pigmentation, and telangiectasias was more significant in the PRP-treated group [32]. These controversial studies show that the lack of standardization may interfere with the quality of the results from using PRP in aesthetics.

Clinical evidence indicates that PRP can be utilized independently or with lasers, microneedling agents, or other substances, such as hyaluronic acid. Combining hyaluronic acid with PRP may amplify the release and retention of growth factors, enhancing collagen synthesis and stimulating fibroblast activation, thereby contributing to skin rejuvenation [31]. A prospective, open-label study by Hersant et al. aimed to showcase the clinical advantages of combining PRP and hyaluronic acid. The study, utilizing FACE-Q scores and biophysical measurements, revealed a substantial improvement at the 6-month mark compared to the baseline [36].

A study carried out in Korea showed that PRP combined with a fractional laser increased patient satisfaction and skin elasticity and decreased the rate of erythema. It also increases the cohesion of the dermal epidermis, the amount of collagen fibers, and the number of fibroblasts [65]. Gawdat et al. compared fractionated radiofrequency (fr-RF) microneedling alone and in combination with PRP for rejuvenating the neck region. The combined treatment showed an overall improvement in the appearance of the neck, as evidenced by the statistically significant increase in the dermal thickness, moderate-to-excellent results according to a medical evaluation by the Global Aesthetic Improvement Scale (GAIS), and great patient satisfaction [14].

PRP and its derivatives associated with other rejuvenation techniques optimize the results and reduce the side effects of some methods, accelerating the regeneration process. One of these associations was demonstrated by Cai et al. [35], who attempted to evaluate whether PRP could improve the restorative effects of erbium fractional laser treatments. This procedure is widely used because it considerably improves skin aging, but it is associated with several side effects, such as erythema and pigmentation. They concluded that erbium fractional laser irradiation combined with PRP application is an effective and safe option for improving aging facial skin, with minimal side effects, suggesting the use of the combination instead of the laser alone.

Atrophic acne scarring presents a therapeutic challenge in aesthetic medicine, but various modalities are available to address this concern. These include microneedling, radiofrequency, fractional lasers, punch excision, suturing, subcision, and dermabrasion, as well as autologous fat grafting, autologous dermal grafts, and autologous PRP and PRF [66]. Cho and colleagues [67] evaluated the effects of PRP in vitro on the activation of dermal fibroblasts through extracellular matrix remodeling. Cell proliferation and migration assays, an ELISA, and Western blotting revealed that PRP elevated the expression of type I collagen, elastin, and matrix metalloproteinases 1 and 2, thereby expediting the wound-healing process. These results offer valuable insights into the potential mechanisms by which PRP facilitates tissue remodeling, suggesting its applicability in addressing aesthetic dysfunctions such as acne scars.

PRP and microneedling have gained popularity as off-label treatments for rejuvenation and body repair. In aesthetic medicine, they represent comprehensive and informed resources. A recent study involving 40 patients with atrophic acne scars compared the efficacy of autologous i-PRF + microneedling to that of microneedling alone. The area treated with i-PRF exhibited a significant reduction in acne scarring, and the mean patient satisfaction score was notably greater [48]. Asif and colleagues [46] conducted a split-face comparative study to treat atrophic acne scars, comparing PRP + microneedling versus microneedling + distilled water. In the PRP + microneedling group, 40% of patients reported “excellent” results, while 60% rated the results as “good”. The study concluded that the combination of PRP and microneedling is significantly more effective than microneedling alone.

Acne scars are among the most common aesthetic dysfunctions that can affect self-esteem and quality of life. Therefore, there is a rising demand for treatment alternatives that can minimize or ameliorate this disorder. An experimental analytical study was conducted on 40 patients to compare PRP administration after subcision for acne scars. The right side of the face was injected with autologous PRP into each scar after performing subcision. The left side of the face was the control, where only subcision was performed. PRP + subcision led to superior results in post-acne scars compared to subcision alone. They concluded that PRP and subcision act synergistically to improve the appearance of acne scars [47].

Diab et al. performed a comparative study using PRF versus PRP and intradermal application and microneedling to treat atrophic acne scars. The participants were allocated into two groups, with each receiving treatments on each side of the face. The left side of the face received the intradermal injection of PRP (group 1) or PRF (group 2), and the right side received topical PRP or PRF followed by microneedling. No significant differences in the acne scar severity were observed on either side of the face. However, the improvement in the PRF group, either alone or combined with microneedling, was significantly greater than that in the PRP-only group [49].

The CO_2_ dot matrix laser is a widely used method for treating acne scars, and its effects have been reported in several studies. However, one of the limitations of this technique is its high cost. Furthermore, the effectiveness of this method depends mainly on the penetration capacity of the laser, the homogeneous energy distribution, and the excessive thermal coagulation, which can cause adverse reactions, such as thermal damage and pigmentation of the patient’s skin [68]. Guo and colleagues studied the clinical efficacy and safety of a CO_2_ dot matrix laser combined with PRP. They concluded that combining the two techniques can strongly improve the scar repair efficacy and psychosocial health and quality of life in patients with acne scars [50].

## 4. Striae

Striae distensae (S.D.) are commonly referred to as stretch marks. S.D. manifest clinically as parallel striae perpendicular to the skin’s tension lines. These tumors resemble dermal scars histologically and are often associated with factors such as pregnancy, obesity, hormonal changes, and genetic predisposition, primarily affecting females. There are two types of stretch marks based on their clinical and histopathological features and categorized by their maturation stages. In the initial stage, stretch marks known as “striae rubra” (S.R.) manifest as immature, tense, and erythematous lesions due to the reorganization of and reduction in elastin and fibrillin fibers and the structural changes in collagen. Over time, they progress to the atrophic and hypopigmented stage and are then referred to as “striae alba” (S.A.), characterized by the local breakdown of elastin and collagen with mast cell enzymes released in the mid-dermal tract [69,70].

Various treatment modalities, including different types of lasers (pulsed-dye, fractional CO_2_, Nd YAG, Er Glass, Er YAG, and diode lasers), radiofrequency, microdermal abrasion, carboxytherapy, chemical peels, and topical cosmetics such as sodium ascorbate and tretinoin, have been explored in numerous studies, yielding variable results for the treatment of stretch marks. However, these methods have not yet been proven to be sufficiently compelling and have minimal adverse effects. Leveraging the well-established biological effects of PRP, this biological tool has emerged as a promising option for treating stretch marks, either as a standalone therapy or in combination with other alternatives [70]. To confirm this hypothesis, de Castro and colleagues [55] subjected patients with abdominal striae to intralesional PRP injections to characterize and compare the structural changes in the collagen and elastic fibers in these areas. Furthermore, the authors suggested that this treatment’s possible mechanisms of action were related to signaling pathways involving Toll-like receptors (TLRs) and growth factors. Biopsies of the treated areas were conducted at the initiation of treatment and at weeks 6 and 12 posttreatment, revealing the effectiveness of PRP in reducing the area of stretch marks. This reduction was accompanied by the stimulation of collagen and elastic-fiber synthesis and remodeling. Moreover, PRP increased the immunoreactivities of TLR2 and TLR4, subsequently increasing the TNF-α, VEGF, and IGF-1 levels. These findings underscore PRP’s promising therapeutic potential for treating stretch marks.

Another recent study described and analyzed stretch-marked-derived fibroblasts (SMFs) subjected to two in vitro treatments: sodium ascorbate and PRP. The type I collagen expression was measured before and after adding different concentrations of PRP and sodium ascorbate to the culture medium. This study demonstrated that SMFs treated with both substances exhibited increases in type I collagen expression and cell proliferation. After 24 h of incubation with 1% PRP or 5% PRP + sodium ascorbate, the cell viability increased by 140% and 151%, respectively, and it increased by 156% and 178%, respectively, after 48 h. These results showed that both treatments were effective and suggested that the improvement in stretch marks mediated by the metabolic activity of SMF was viable [71].

In light of the information above, it is evident that the utilization of PRP can play a crucial role in treating stretch marks. This process appears to be mediated by the release of growth factors that act on the proliferation of fibroblasts, accounting for tissue repair and the production of collagen and elastic fibers. Based on these hypotheses, Abdel-Motaleb [53] assessed whether incorporating PRP would enhance the efficacy of microneedling in addressing stretch marks. Forty individuals with stretch marks were separated into two groups: those receiving microneedling alone or those receiving microneedling in conjunction with PRP. The study findings indicated that the combined therapy resulted in the superior enhancement of skin lesions, improved collagen and elastic-fiber deposition, heightened fibroblast proliferative activity, and the reduced expression of the caspase-3 protein in the epidermis.

Various kinds of lasers, such as fractional CO_2_ lasers, are effective for S.D. To evaluate the synergistic effect of this laser alone with PRP in S.D. treatment, Sayed et al. [56] compared the efficacy of a fractional CO_2_ laser alone versus a CO_2_ laser + PRP in S.D. treatment. They studied this effect in thirty adult female patients with S.D. divided into two groups: laser monotherapy or laser + PRP. Skin biopsies were taken from the lesions before and after treatment for histopathological evaluation. The authors observed that the combined treatment was more effective than the fractional CO_2_ laser alone. Similarly, Neinaa and colleagues [51] evaluated the synergistic effect of a fractional CO_2_ and pulsed-dye laser (PDL) with PRP in thirty S.D. patients. Patients received an intradermal injection of PRP on both sides, followed by a fractional CO_2_ laser on the right side and a PDL on the left side. The patients received three treatment sessions at 6-week intervals. The authors observed that both treatment sides had significantly improved clinical S.D. lesion outcomes; a significantly greater degree of clinical improvement with better outcomes and fewer side effects were observed in response to the PRP + fractional CO_2_ laser irradiation than in response to the PRP + PDL.

However, Preclaro and colleagues [52] also studied the synergistic effect of a fractional CO_2_ laser with PRP, albeit in striae gravidarum (S.G.), a connective-tissue dysfunction commonly observed in primigravidae. They evaluated 16 patients with S.G. treated with a fractional CO_2_ laser followed by PRP on one side of the abdomen and with a fractional CO_2_ laser followed by normal saline on the other. The study was performed in three sessions at 4-week intervals and showed that the combination of an ablative fractional CO_2_ laser and autologous PRP was superior regarding the clinical improvement and patient satisfaction score, but the outcome measures were not significantly different. These findings may suggest that differences in the protocols or types of striae may interfere with the synergistic effects of PRP and other treatments.

Ebrahim et al. [54] compared the efficacy and safety of PRP monotherapy versus PRP + subcision or medium-depth peeling (70% glycolic acid followed by 35% trichloroacetic acid) in seventy-five female patients with S.D. They concluded that using PRP in combination with subcision or peeling was more effective than using PRP alone.

## 5. Authors’ Note: Choosing between PRP and i-PRF for Aesthetic Procedures

Physicians often face the dilemma of selecting the most suitable platelet derivative for aesthetic procedures. Recent studies have shed light on the comparative effectiveness of PRP and i-PRF in promoting skin rejuvenation and enhancing aesthetic outcomes.

One study [72] demonstrated that i-PRF stimulates greater dermal skin fibroblast cell migration and proliferation and collagen synthesis compared to PRP. The study found that all platelet concentrates were non-toxic to cells and promoted high cell survival rates. Fluid-PRF showed significant advantages over the PRP and control groups: skin fibroblasts migrated over 350% more compared to the control and 200% more compared to the PRP group, induced greater cell proliferation at 5 days, and resulted in significantly higher mRNA levels of TGF-beta, collagen 1, and fibronectin compared to the PRP group. Additionally, fluid-PRF demonstrated a significantly greater ability to induce collagen matrix synthesis than PRP. This finding underscores the potential superiority of i-PRF in promoting tissue regeneration and enhancing aesthetic results.

Moreover, a comprehensive review [7] highlights an important consideration regarding the diffusibility of PRP and the durable scaffolding effect of fibrin from PRF. The therapeutic value of PRP, known for its diffusible nature, may sometimes be hindered depending on the injection site. In contrast, i-PRF offers a natural product with no anticoagulant needed, minimizing the interference with natural biochemistry and lowering costs. However, i-PRF does require a more experienced operator due to the technical complexities in its preparation. The operator must collect blood quickly and centrifuge it immediately to avoid premature coagulation, ideally within 2 min and 30 s [7]. Despite these challenges, i-PRF presents a more feasible treatment alternative, particularly in procedures requiring enhanced tissue regeneration and longevity.

In light of these findings, physicians should carefully consider the specific requirements of each aesthetic procedure and the desired outcomes when choosing between PRP and i-PRF. While PRP may offer potent signaling effects, i-PRF’s durable scaffolding effect and superior cell stimulation properties, along with its feasibility and cost-effectiveness, make it a promising option for achieving optimal aesthetic results.

Ultimately, the decision between PRP and i-PRF should be guided by a thorough understanding of their properties, mechanisms of action, and clinical evidence, in conjunction with the individual needs and goals of each patient.

## 6. Conclusions

While the literature underscores the potential benefits of PRP and PRF in addressing various aesthetic concerns, such as wrinkles, skin texture, hair loss, acne scars, and hyperpigmentation, challenges persist within the field. The lack of standardized protocols for PRP and PRF preparation remains a significant impediment, leading to variability in the methods and concentrations across studies and hindering consistency and efficacy.

The absence of an official PRP protocol for precise goals is due to various factors, primarily the lack of standardization in PRP preparation methods. Variability in techniques, such as centrifugation and anticoagulants, leads to diverse platelet concentrations and growth factor compositions, hindering uniform treatment outcomes. Additionally, the diverse applications of PRP across medical specialties require tailored protocols. Different treatment objectives, such as orthopedic versus dermatological applications, necessitate specialized formulations and delivery methods. However, consensus on optimal PRP formulations remains elusive due to the complex interplay of variables.

Despite these hurdles, PRP and PRF offer promising, minimally invasive treatments for aesthetic concerns. However, the reliance on small sample sizes in some studies underscores the challenges in drawing robust conclusions from the available evidence. While the evidence suggests a promising role for PRP and PRF, it is essential to acknowledge the limitations inherent in the current body of literature. Moving forward, efforts to address these challenges will be crucial in advancing the field and ensuring the reproducibility and reliability of outcomes. To establish standardized PRP protocols, collaborative efforts among stakeholders are crucial. Consensus guidelines, informed by empirical evidence from clinical trials, are needed to address this challenge. Technological advancements, such as automated processing systems, may offer potential solutions to enhance standardization.

In conclusion, while PRP and its derivatives hold significant promise as minimally invasive treatments for various aesthetic concerns, addressing the current challenges and uncertainties surrounding their use is paramount. By fostering collaboration and driving innovation, the aesthetic medicine community can overcome these challenges and realize the full potential of PRP and i-PRF in enhancing patient outcomes and satisfaction.

## Figures and Tables

**Figure 1 ijms-25-05604-f001:**
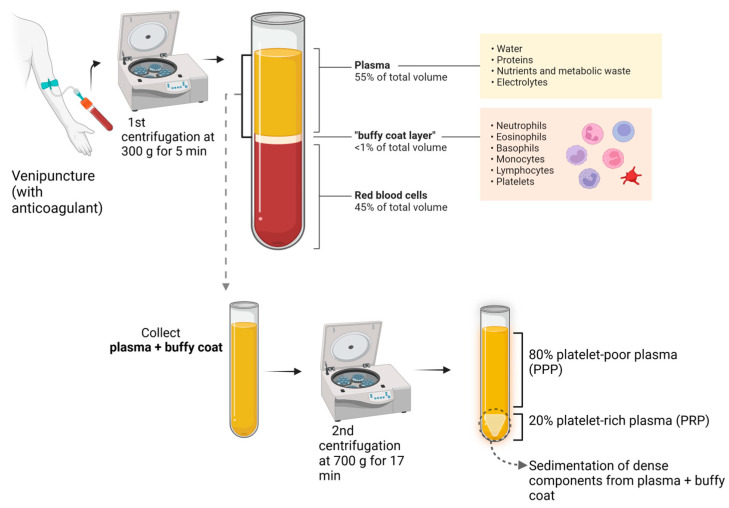
PRP preparation.

**Figure 2 ijms-25-05604-f002:**
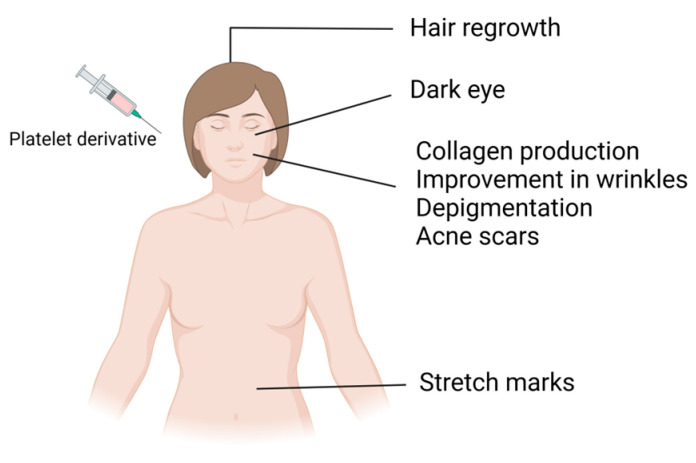
Application of platelet derivatives in regenerative aesthetics.

**Figure 3 ijms-25-05604-f003:**
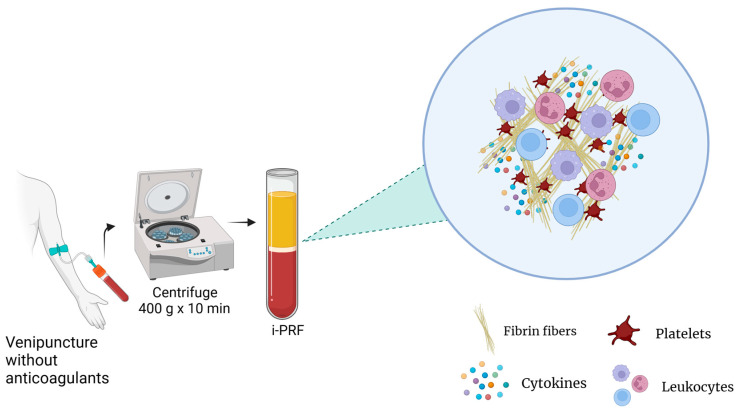
PRF preparation.

**Table 1 ijms-25-05604-t001:** Baseline characteristics of the randomized clinical trials included in the studies.

Reference	Application	Number of Patients	Delivery Methods	Treatment Schedule	Results	Clinical Significance
Trink, A.; et al. (2013) [16]	Alopecia areata	45 participants	PRP intradermal injection	3× monthly	Increased hair regrowth and decreased hair dystrophy and burning or itching sensation.	Highly significant
Vazques, O.A.; et al. (2022) [17]	Alopecia areata	1 participant	PRF intradermal injection	2× monthly	Complete resolution of alopecia areata at 6-month follow-up.	Highly significant
Gentile, P.; et al. (2015) [18]	Alopecia androgenetica	20 participants	PRP intradermal injection	3× monthly	Improvement in the mean number of hairs, with a mean increase of 33.6 hairs, and increase in total hair density of 45.9 hairs per cm^2^.	Significant
Alves, R. and R. Grimalt (2016) [19]	Alopecia androgenetica	25 participants	PRP intradermal injection	3× monthly	Improvement in anagen hairs, telogen hairs, hair density, and terminal hair density.	Significant
Gentile, P.; et al. (2018) [20]	Alopecia androgenetica	23 participants	PRP intradermal injection	3× monthly	An increase of 33.6 hairs in the target area and a mean increase in density of 45.9 hairs per cm^3^ compared with placebo.	Significant
Butt, G.; et al. (2019) [21]	Alopecia androgenetica	30 participants	PRP intradermal injection	2× monthly	Mean hair density on first visit (before treatment) was 34.18 ± 14.36/cm^2^, which was increased to 50.20 ± 15.91/cm^2^ 6 months after first treatment. Mean scores of physician and patient global assessments were 1.45 ± 0.57 and 1.60 ± 0.62, respectively. Mean percentage reduction in hair pulled was 29.2%. Terminal-to-vellus hair ratio was increased in 60% of patients.	Highly significant
Rodrigues, B.L.; (2019) [22]	Alopecia androgenetica	26 participants	PRP intradermal injection	4× monthly	Increases in hair count, hair density, and percentage of anagen hairs in the PRP group versus in the control group, without correlation with platelet counts or quantification of the growth factors in PRP.	Significant
Shapiro, J.; et al. (2020) [23]	Alopecia androgenetica	35 participants	PRP intradermal injection	3× monthly	There was no significant difference in hair density change between the PRP group and placebo group.	Useful
Qu, Q.; et al. (2021) [13]	Alopecia androgenetica	Mice models and 52 participants	PRP intradermal injection	6× monthly	PRP treatment boosted hair regrowth and accelerated hair cycling, and the effect was sustained for more than one hair cycle in mice. Clinically, mean hair count, density, diameter, and anagen hair ratio in PRP group showed significant improvements compared to control side.	Highly significant
Moftah, N.H.; et al. (2022) [24]	Female pattern hair loss	40 participants	PRP intradermal injection	3× monthly	Increases in percentage of terminal hair and average width of hair after treatment.	Significant
Muhammad, A.; et al. (2022) [25]	Alopecia androgenetica	60 participants	PRP microneedling versus PRP intradermal injection	3× monthly	Patients in the microneedling group achieved a negative hair-pulling test and had improved perception of hair loss compared to the PRP-alone group (82.1% vs. 51.9% and 88.0% vs. 73.9%, respectively). The percentage increase in the mean hair count in the microneedling group (24.53 ± 9.49%) was significantly higher than the increase in the PRP-alone group (17.88 ± 10.15%) (*p* = 0.011).	Highly significant
Ozcan, K.N.; et al. (2022) [26]	Alopecia androgenetica	62 participants	PRP microneedling versus PRP intradermal injection	Three sessions at 2-week intervals and the fourth session 1 month after the last session	Hair-pulling test became significantly negative after treatment (*p* < 0.05). Improvements in hair count, hair density, terminal hair count, and terminal hair density in both groups compared to pretreatment (*p* < 0.05). Between the groups, a statistically significant difference was found between the averages of anagen hair, telogen hair, and hair length in the microneedling-treated group compared to the group treated with the point-by-point technique.	Highly significant
Wu, S.; et al. (2023) [27]	Alopecia androgenetica	75 participants	PRPF intradermal injection versus 5% topical minoxidil versus PRF injection combined with minoxidil.	Topical minoxidil at 5% twice daily; PRPF injection performed three times, 1 month apart	All patients showed improvements in hair count and terminal hair and decreases in telogen hair ratio after treatment. The efficacy of PRF complex therapy revealed significant improvements in hair count, terminal hair, and growth rate, compared with monotherapy.	Highly significant
Wei, W.; et al. (2023) [28]	Alopecia androgenetica	30 participants	PRP intradermal injection combined with 5% topical minoxidil versus PRP injections	PRP treatment thrice monthly; minoxidil at 5% twice a day for 3 months	Significant increases in all patients in hair density and quantity after PRP treatment, and there was no significant difference in mean hair diameter. Hair density/quantity was more pronounced in group with PRP injections combined with 5% topical minoxidil than in group with PRP alone.	Highly significant
El-Dawla, R.E.; M. Abdelhaleem, and A. Abdelhamed (2023) [29]	Telogen effluvium	30 participants	PRP intradermal injection	4× monthly	The hair-pulling test, visual analog scale, and patient satisfaction results showed statistically significant differences in PRP group.	Significant
Elnehrawy, N.Y.; et al. (2017) [30]	Skin rejuvenation (wrinkles)	20 participants	PRP intradermal injection	Single PRP intradermal injection	The mean value of the wrinkle severity was reduced from 2.90 ± 0.91 before treatment to 2.10 ± 0.79 after 8 weeks of treatment. The most significant results were with younger subjects that had mild and moderate wrinkles of the nasolabial folds (NLFs).	Significant
Hersant, B.; et al. (2017) [31]	Skin booster (PRP + hyaluronic acid)	31 participants	Mix of PRP and hyaluronic acid intradermal injection versus microneedling	3× monthly	FACE-Q scores showed improvement at 6 months compared with baseline (44.3 ± 1.9 at baseline versus 52 ± 3.17 at 6 months). In addition, there was an improvement in the net elasticity parameter (*p* = 0.036) from baseline.	Highly significant
Alam, M.; et al. (2018) [32]	Skin rejuvenation (cheek rhytids)	27 participants	PRP intradermal injection	Single PRP injection	PRP-treated side had significant improvements compared to normal saline for texture (mean self-assessment score, 2.00 [1.20] vs. 1.21 [0.54]) and wrinkles (mean self-assessment score, 1.74 [0.99] vs. 1.21 [0.54]).	Significant
Everts, P.A.; P.C. Pinto, and L. Girao (2019) [33]	Skin rejuvenation (brown spots, skin redness, and firmness)	11 participants	PRP intradermal injection	3× monthly	Decrease in brown-spot counts and areas. Wrinkle count and volume were significantly reduced. Skin firmness and redness were significantly improved. The self-assessment at 6-month follow-up revealed an average satisfaction score of >90%.	Highly significant
Hassan, H.; D.J. Quinlan, and A. Ghanem (2020) [34]	Skin rejuvenation (texture, wrinkles, ultraviolet spots, and porphyrins)	11 participants	PRF intradermal injection	3× monthly	Improvement in skin surface spots (*p* = 0.01), pores (*p* = 0.03), skin texture, wrinkles, ultraviolet spots, and porphyrins. Satisfaction with appearance values all showed significant improvements from baseline, including satisfaction with skin (*p* = 0.002), satisfaction with facial appearance (*p* = 0.025), satisfaction with cheeks (*p* = 0.001), satisfaction with lower face and jawline (*p* = 0.002), and satisfaction with lips (*p* = 0.04).	Highly significant
Cai, J.; et al. (2020) [35]	Skin rejuvenation	158 participants	Erbium fractional laser and topical PRP combination	Two sessions 2 months apart	The symptoms of skin aging, especially skin color, pore expansion, and skin texture, showed improvements, according to the evaluation of the patients and the physicians of 90.51% and 88.61%, respectively.	Significant
Hersant, B.; et al. (2021) [36]	Skin booster	93 participants	PRP, hyaluronic acid, and a mixture of PRP and H.A. intradermal injection and micrconeedling	3× monthly	Treatment with mixture of PRP and H.A. led to a very significant improvement in the overall facial appearance compared with treatment with PRP or H.A. alone (*p* < 0.0001). Biophysical measurements also showed significantly improved skin elasticity in this group.	Highly significant
da Silva, L.Q.; et al. (2021) [37]	Skin rejuvenation	19 participants	Lyophilized PRP intradermal injection	2× monthly	The use of lyophilized PRP by mesotherapy showed no improvement in skin aging.	Significant
Banihashemi, M.; et al. (2021) [38]	Skin rejuvenation (periorbital dark circles and nasolabial folds)	30 participants	PRP intradermal injection	Two sessions with a 3-month interval	Improvement was statistically significant for dark circles and nasolabial folds (*p* value 0.025).	Highly significant
Nilforoushzadeh, M.A.; et al. (2021) [39]	Skin rejuvenation (periorbital hyperpigmentation and wrinkles)	32 participants	Er: YAG laser and PRP intradermal injection or topical combination versus Er: YAG laser alone.	3× monthly	The periorbital melanin in the combined group was significantly decreased. Significant increase in the skin lightness and decrease in the percent change of the color and wrinkles in the combined group.	Highly significant
Mumtaz, M.; et al. (2021) [40]	Melasma	64 participants	PRP intradermal injection versus tranexamic acid intradermal injection	Fourth week and for a total period of 12 weeks	Intradermal PRP was significantly better than intradermal tranexamic acid in management of melasma.	Useful
Diab, H.M.; et al. (2022) [41]	Skin rejuvenation (periorbital wrinkles)	40 participants	PRP intradermal injection versus plasma gel	Two treatment sessions 4 weeks apart	Both modalities yielded a significant improvement in periorbital wrinkles after the second session, with significantly better results on the plasma gel-injected side.	Significant
Gawdat, H.; et al. (2022) [14]	Neck rejuvenation	20 participants	Fractional radiofrequency microneedling versus in combination with PRP	3× monthly	Both groups showed statistically significant improvements in all parameters. Comparing the two groups, the mean dermal thickness after treatment was higher in the combined group.	Highly significant
Patil, N.K. and A.K. Bubna (2022) [42]	Melasma	40 participants	Tranexamic acid (TXA) versus PRP intradermal injection	Once every 4 weeks for a total of five treatment sessions	Both TXA and PRP were found to be effective and safe for melasma, providing rapid and substantial improvement even when used as standalone therapies, although PRP was found to be slightly better than intradermal TXA.	Highly significant
Gonzalez-Ojeda, A.; et al. (2022) [43]	Melasma	20 participants	PRP intradermal injection	3× biweekly	PRP was associated with decreased intensity of the melasma patch and improved skin quality.	Highly significant
Shen, J.; et al. (2023) [44]	Skin booster (coarse pores and wrinkles)	90 participants	PRP intradermal injection	8× monthly	PRP treatment significantly improved (*p* < 0.05) the patients’ facial skin indicators, quality of life, and satisfaction.	Highly significant
Mahmoodabadi, R.A.; et al. (2023) [45]	Periorbital wrinkles	16 participants	PRF matrix subdermis injection using a canola	1 session	Noticeable improvements in deep, fine, and small wrinkles, periocular hyperpigmentation, and overall skin freshness at the injection site.	Significant
Asif, M.; S. Kanodia, and K. Singh (2016) [46]	Atrophic acne scars	50 participants	PRP intradermal injection and combined with microneedling	3× monthly	PRP has efficacy in the management of atrophic acne scars. It can be combined with microneedling to enhance the final clinical outcomes in comparison with microneedling alone.	Significant
Deshmukh, N.S. and V.A. Belgaumkar (2019) [47]	Atrophic acne scars	40 participants	PRP injection into acne scar after performing subcision	Four sessions on each side of the face with an interval of 4 weeks between two consecutive sessions	Platelet-rich plasma and subcision showed greater improvement (32.08%) in post-acne scars as compared to subcision alone (8.33%). Rolling acne scars responded greatest (39.27%), followed by box-type scars (33.88%).	Highly significant
Krishnegowda, R.; S.N. Pradhan, and V.A. Belgaumkar (2023) [48]	Atrophic acne scars	40 participants	PRF intradermal injection followed by microneedling	4× monthly	Acne scars were significantly reduced (1.47, SD 0.56) in comparison to the control side (3.33, SD 0.53). Patient satisfaction score was significantly higher on PRF side (5.95).	Highly significant
Diab, N.A.F., A.M. Ibrahim, and A.M. Abdallah (2023) [49]	Atrophic acne scars	30 participants	PRF and PRP intradermal injections versus PRF and PRP combined with microneedling	Four sessions with 3-week interval	A significant improvement was seen in both PRF and PRP treatment groups. The therapeutic response was significantly higher in the PRF group rather than PRP either alone or combined with microneedling. The combination with needling increases the efficacy of PRF and PRP.	Highly significant
Guo, R.; et al. (2023) [50]	Atrophic acne scars	81 participants	PRP combined with CO_2_ dot matrix laser	3× monthly	CO_2_ dot matrix laser combined with PRP can strongly improve the clinical efficacy on patients and shorten the scar-scabbing time and decrustation time, more effectively contributing to the scar repair, comfort, skin condition, psychological state and satisfaction, and pain reduction.	Highly significant
Neinaa, Y.M.E., S.F. Gheida, and D.A.E. Mohamed (2021) [51]	Stretch marks (striae rubra and/or striae alba)	30 participants	PRP intradermal injection followed by fractional CO_2_ laser or pulsed-dye laser.	Three treatment sessions at 6-week intervals	Significantly higher degrees of clinical improvements were observed in response to treatment sessions by combining PRP with a Fr CO_2_ laser rather than combining PRP with PDL.	Highly significant
Preclaro, I.A.C., E.A.V. Tianco, and M. Buenviaje-Beloso (2022) [52]	Stretch marks (striae gravidarum)	16 participants	PRP combined with fractional carbon dioxide (CO_2_) laser	3× monthly	The combination of an ablative fractional CO_2_ laser and autologous PRP had better clinical improvements and patient satisfaction compared with an ablative fractional CO_2_ laser and placebo.	Highly significant
Abdel-Motaleb, A.A.; et al. (2022) [53]	Stretch marks (striae rubra and/or striae alba)	40 participants	PRP combined with microneedling	3× monthly	PRP with microneedling was associated with an improvement in the skin lesions of striae, a more significant deposition of collagen and elastic fibers, increased proliferative activity in the epidermis, and decreased caspase-3 protein expression values in the epidermis.	Highly significant
Ebrahim, H.M.; et al. (2022) [54]	Stretch marks (striae rubra and/or striae alba)	75 participants	PRP intradermal injection, PRP + subcision, and PRP + combined peeling (GA 70% + TCA 35%).	6× monthly	Significant decreases in all striae measurements were achieved with all groups (*p* < 0.001); however, the combined groups showed greater decreases (*p* = 0.2 and 0.4). All groups demonstrated improvements in dermal collagen deposition, which was higher in the combined groups.	Significant
de Castro Roston, J.R.; et al. (2023) [55]	Stretch marks (striae rubra and/or striae alba)	12 participants	PRP intralesional injection	1× weekly over 12 weeks	PRP treatment was most effective at reducing the area of stretch marks, with consequent stimulation of the synthesis and remodeling of collagen and elastic fibers. Additionally, PRP promoted an increase in TLR2 and TLR4 immunoreactivities, with consequent increases in TNF-α, VEGF, and IGF-1.	Significant
Sayed, D.S.; et al. (2023) [56]	Stretch marks (striae rubra and/or striae alba)	30 participants	PRP intradermal injection combined with fractional CO_2_ laser.	3× monthly	Significant improvements in stretch marks were achieved in the group combining the two techniques (*p* = 0.007), and the average improvement was significantly greater (60.33 ± 26.49) in this group (43.80 ± 27.43) (*p* value = 0.001).	Highly significant

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
