# Peer review of "The Biological Role of Platelet Derivatives in Regenerative Aesthetics"

_ijms, 2024, doi:10.3390/ijms25115604_

Round 1

Reviewer 1 Report

Comments and Suggestions for Authors

This is an interesting review on PRP topic.

I would be beneficial to specify that it is a narrative review, without statistical analysis.

I would use a paragraph to show the different PRP formulations used in aesthetic medicine.

For alopecia and other aesthetic treatments I would cite and put overall calculated numbers of the most cited meta-analysis.

Limitations of this review, as well as future perspective should be better explained.

Author Response

Dear reviewer, thank you for analyzing our manuscript.

We are delighted to know that you found our manuscript interesting.

At the very top left of the manuscript it says that our work is a narrative review.

We have rewritten sections of the manuscript for improvements.

Table 1 contains several clinical trials with more details, including alopecia. Table 1 has been modified and now contains one more column categorizing each study as “clinically useful”, or “significant”, or “highly significant”.

Towards the end of the paper, we have added another section titled “author’s note” where we add more explanations. The conclusion has also been completely rewritten with a forward look on future directions.

Reviewer 2 Report

Comments and Suggestions for Authors

The Review article by and coworkers deals with the uses of PRP in aesthetic medicine. The review has been written with method, after a careful literature search. In this reviewer's humble opinion, two major points and some minor issues should be solved to improve the quality of the work.

Further details are reported below.

Major points:

1. The Introduction section should provide a state-of-art of current literature on the uses of PRP and PRF in regenerative aesthetics. In a review addressed to experts in this field, the detailed procedure to prepare PRP is redundant. Is should be removed, or replaced by a reference. Focusing on the impact of different protocols on PRP quality is definitely more interesting, as the Authors did in lines 75-96. 

2. The Conclusions section cannot end only reporting the need for further studies. The Authors should edit this section, adding their critical point-of-view, answering the main questions of potential readers, providing an expert outlook on the future challenges in this research field. As an example, the questions of a reader of such a review could be: 

- What are the main current challenges in this field?

- Why is not there an official protocol to obtain PRP for precise goals?

- What does it hinders this achievement? How PRP should be different, based on its exploitation?

Minor issues:

3. Table 1 takes several pages and interrupts the reading of the review. It could be reported as an appendix or as a Supplementary material.

4. Please check the spelling in Figure 1 (i.e. "volumel")

5. Section 3.2 Rejuvenation should be renamed as "Skin rejuvenation"

6. Section 4 is missing. Please, update the section numbering.

7. In the Conclusions section, please edit the first sentence (lines 550-552), which is inconclusive.

8. Author contributions should be written exploiting the CRediT taxonomy 

9. Please check the citation format, in particular the use of bold and italics.

Comments on the Quality of English Language

English language does not hinder the understandability of the text. However, the whole review should be checked for typos and inconclusive sentences.

Author Response

Dear reviewer, please see attachment containing replies to your comments.

Reviewer 3 Report

Comments and Suggestions for Authors

This review is an update summary on the properties and biomedical application of platelet derivatives in regenerative aesthetics. The review is fine, well-written and useful for the field. Especially Table 1 (randomized clinical trials) is a valuable source of information for future investigators. I have 2 suggestions which may improve the readability of the paper.

1) A graphic figure on the platelet derivatives/properties reviewed here would be helpful.

2) The authors should group the many trials reviewed here (Table 1) in catagories such as "Highly significant", "significant", "useful".     

Comments on the Quality of English Language

see above

Author Response

Dear reviewer, please see attachment.

Round 2

Reviewer 1 Report

Comments and Suggestions for Authors

All suggestions have been mostly fulfilled.

Reviewer 2 Report

Comments and Suggestions for Authors

The Reviewer's questions have been addressed